# Impact of Coastal Walking Outdoors and Virtual Reality Indoor Walking on Heart Rate, Enjoyment Levels and Mindfulness Experiences in Healthy Adults

**DOI:** 10.3390/jfmk9010011

**Published:** 2023-12-29

**Authors:** Gianpiero Greco, Claudio Centrone, Luca Poli, Ana Filipa Silva, Luca Russo, Stefania Cataldi, Valerio Giustino, Francesco Fischetti

**Affiliations:** 1Department of Translational Biomedicine and Neuroscience (DiBraiN), University of Study of Bari, 70124 Bari, Italy; gianpiero.greco@uniba.it (G.G.); claudio.centrone@uniba.it (C.C.); luca.poli@uniba.it (L.P.); francesco.fischetti@uniba.it (F.F.); 2Sports and Leisure School, Polytechnic Institute of Viana do Castelo, 4900-347 Viana do Castelo, Portugal; anafilsilva@gmail.com; 3Research Center in Sports Performance, Recreation, Innovation and Technology (SPRINT), 4960-320 Melgaço, Portugal; 4The Research Centre in Sports Sciences, Health Sciences and Human Development (CIDESD), 5001-801 Vila Real, Portugal; 5Department of Human Sciences, Università Telematica degli Studi IUL, 50122 Florence, Italy; l.russo@iuline.it; 6Sport and Exercise Sciences Research Unit, Department of Psychology, Educational Science and Human Movement, University of Palermo, 90144 Palermo, Italy; valerio.giustino@unipa.it

**Keywords:** VR, technology, physical activity, physical fitness, exercise adherence

## Abstract

Outdoor exercise is beneficial for psychophysical well-being. Limited studies have compared outdoor and virtual reality (VR) indoor physical activities, especially in coastal settings. Therefore, this study aimed to assess the impact of outdoor coastal walking and indoor walking in a VR simulation with a similar environment on physiological and psychological variables in healthy adults. A total of 26 subjects (14 M and 12 F, age 25.2 ± 2.5 years) voluntarily participated in this crossover randomized controlled and counterbalanced study and were allocated under three conditions: VR indoor walking (INVR), outdoor walking (OUT) and standard indoor walking (IN). IN and INVR conditions were performed on a treadmill (speed 4.5 km/h) and the OUT was performed on a seaside pedestrian road. The same outdoor environment was displayed in the visor during the INVR. Heart rate (HR_mean/max_), physical activity enjoyment (PACES-It) and state of mindfulness for physical activity (SMS-PA) were assessed at the end of each condition. The OUT condition showed significantly greater PACES-It scores and HR_mean_ than IN and INVR (*p* < 0.001) and greater SMS-PA scores and HR_max_ than IN (*p* < 0.01 and *p* < 0.05, respectively). No significant differences were found between OUT and INVR regarding HR_max_ and SMS-PA scores (*p* > 0.05). Findings suggest that physical activity in an immersive technology may lead to physiological loads comparable to the outdoor environment. OUT is more enjoyable than IN and INVR but exhibits a mindfulness response comparable to INVR. Therefore, INVR could be an alternative to OUT for those who cannot engage in outdoor activities for various reasons.

## 1. Introduction

In the aftermath of the COVID-19 pandemic, the global workforce witnessed a substantial transition towards remote and digital work models. However, this shift has inadvertently given rise to a concerning surge in sedentary behaviors, posing a significant threat to individual well-being [1]. As the boundaries between personal and professional spaces blur, people increasingly find themselves immersed in prolonged periods of screen time, often seated for extended hours. This sedentary lifestyle has been associated with various health issues, including musculoskeletal problems, obesity, and cardiovascular concerns [1]. A sedentary lifestyle, characterized by prolonged periods of sitting and a decline in physical activity (PA) levels, is intricately linked to psychological and social well-being. Scientific evidence indicates that sedentary behavior is associated with an increased risk of mental health issues, including heightened stress, anxiety, and depression [2]. Furthermore, the social ramifications of a sedentary lifestyle are notable. Reduced physical activity can contribute to social isolation [3] and a diminished quality of life, affecting interpersonal relationships and overall social engagement. Consequently, addressing this emergent challenge requires a comprehensive approach that not only acknowledges the convenience of remote work but also actively promotes PA and ergonomic practices to safeguard the health and well-being of the workforce in this evolving professional landscape.

The surrounding environment, particularly the natural outdoors, has proven to play a vital role in improving relaxation levels and enhancing the overall enjoyment of PA [4]. Connecting with nature and engaging in outdoor exercises not only helps individuals break away from the monotony of indoor settings but also provides a refreshing and invigorating experience that positively impacts their mental state [5]. Researchers have found that being in nature can significantly reduce stress levels, improve mood, and boost cognitive function [6]. PA, therefore, becomes a crucial factor in making the most of the outdoor natural environment’s benefits [7]. Engaging in various outdoor activities, be they hiking, cycling, running, or simply taking a walk in the park, brings numerous advantages across different aspects of an individual’s life. A previous study has shown that outdoor activities have a positive impact on the social, psychological, and physiological well-being of individuals [7]. Not only do these activities promote physical health, but they also contribute to a sense of community and connectedness as people often engage in outdoor exercises with friends, family, or in group settings [8]. Moreover, outdoor activities have demonstrated preventive effects in both young and elderly populations, helping to reduce the risk of various health conditions [7].

An aspect insufficiently explored within expectancy-value theories linked to PA is enjoyment, synonymous with intrinsic motivation [9]. Enjoyment represents a positive emotional state encompassing sensations like pleasure, liking, and fun [10]. Previous studies, both correlational and descriptive, have suggested a potential association between enjoyment and youth engagement in PA [9,11]. It is known that social interactions, either directly or indirectly, enhance enjoyment, especially in entertainment services like online games [12]. As the game and entertainment industries dominate virtual reality (VR), increased social interactions among VR users are anticipated to boost enjoyment and usage intentions [13].

Mindfulness is an important factor for enhancing PA as well: in fact, there is a positive relationship between dispositional mindfulness and PA, particularly with psychological factors related to PA [14]. Mindfulness-based interventions demonstrated a greater probability of effectiveness when specifically customized for PA, targeting psychological aspects linked to engaging in physical exercises [14]. Elevated levels of mindfulness during PA could aid individuals in addressing challenges related to self-control, self-regulation, and body image concerns. This, in turn, may amplify internal motivation, contributing to sustained PA and heightened trait mindfulness. Consequently, it is crucial to assess mindfulness during PA and utilize dependable and valid measurement instruments for this purpose [15].

As technology continues to advance, innovative solutions have emerged to tackle sedentary behavior and motivate people to lead active lifestyles. Exergames, a combination of exercise and gaming, have gained popularity due to their effectiveness in increasing motivation and changing sedentary habits [16,17,18]. These interactive games often require physical movements, making exercise more engaging and enjoyable for individuals [17]. Furthermore, the integration of augmented reality (AR) and VR in physical exercise has shown promising results in terms of enhancing the overall exercise experience [19]. AR and VR technologies offer immersive and interactive workout environments that capture users’ attention and provide novel ways to engage in PA [17]. AR and VR represent distinct immersive technologies. While VR creates entirely artificial environments, isolating users from the real world, AR overlays digital information onto the real world, enhancing the user’s perception. VR often involves dedicated headsets, offering deep immersion, whereas AR can be experienced through various devices, blending digital content with the real environment [20].

A previous study has shown that exercising with AR and VR has positive effects on both PA levels and psychological well-being [19]. The virtual settings can make workouts more interesting and challenging, encouraging individuals to participate more frequently and maintain their exercise routines [19]. In particular, virtual reality has shown significant potential to improve exercise outcomes due to an increase in training session frequency and has been found to enhance muscular strength [19]. Moreover, the use of VR has demonstrated promising results in reducing chronic pain for certain individuals, making it a valuable tool in rehabilitation and pain management [21].

As of today, only a few studies have examined the difference between outdoor activities and indoor activities that incorporate AR and VR technologies [22,23], and none of them have specifically explored the coastal environment. Therefore, this study aimed to assess the impact of outdoor coastal walking (OUT) and indoor walking in a VR simulation (INVR) with a similar environment on some physiological (i.e., heart rate) and psychological (i.e., enjoyment levels and mindfulness experiences) measures in healthy adults. We hypothesized that all measured variables of the INVR condition would have values similar to the OUT condition.

## 2. Materials and Methods

### 2.1. Participants

A total of 26 healthy adults (age, 25.2 ± 2.5 years; body mass, 67.7 ± 8.6 kg; body height, 171.0 ± 10.0 cm; BMI, 23.1 ± 2.4 kg/m^2^; gender, 12 females and 14 males) voluntarily participated in the study carried out in June 2023. Participants were recruited from the University of Bari (Italy) and included students in the bachelor’s degree and master’s degree courses in Sports Science aged between 19 and 30 years. The exclusion criteria were as follows: (i) refusal to participate in the study, (ii) symptoms or signs of musculoskeletal disorders or other severe lower extremity injuries, (iii) presence of acute or chronic disease, and (iv) failure to attend the study protocol.

To establish the sample size needed for the study, an a priori power analysis [24] with an assumed type I error of 0.05 and a type II error rate of 0.20 (80% statistical power) was calculated and revealed that 13 participants in total would be sufficient to observe medium effect sizes “within-subjects”.

Before the study, the participants signed the informed consent document, which provided detailed explanations of the activities and tests that would be administered during the study and the possibility of retiring at any time. This study was conducted in accordance with the Declaration of Helsinki and approved by the Ethics Committee of Bari University (protocol code 0015637|16 February 2023).

### 2.2. Study Design

A randomized controlled crossover study was used (within-subjects repeated-measures design). This study design was employed to assess the acute effects of each walking condition on participants’ physiological and psychological variables. Each subject underwent each condition in a random and counterbalanced order.

Participants were randomly assigned to three walking conditions: (1) indoor walking with virtual reality (INVR) in the coastal environment, (2) outdoor coastal walking (OUT), or (3) standard indoor walking (IN). The randomization was performed by Research Randomizer, a program published on a publicly accessible official website (www.randomizer.org, accessed on 1 June 2023). The three conditions and all assessments were conducted at the same time of the day to minimize possible circadian-related effects—between 9 a.m. and 12 a.m. with a 2-day washout period between trials. This timeframe was chosen to minimize the impact of external temperatures during the OUT session. Moreover, the subjects were asked to avoid strenuous PA and caffeine intake in the 24 h preceding each condition and during the data collection.

Participant characteristics and all outcome measures obtained after each walking condition were assessed by Researcher 1, who was blinded to treatment allocations. The interventions and assessments were performed in the same coastal environment and indoor gym at the University Sports Center in Bari (Italy). Researcher 2 conducted the interventions and was not involved in the subject assessment. Both researchers were instructed not to communicate with subjects about study goals or treatments.

Figure 1 shows a flow chart of the randomized allocation of participants to the three conditions.

### 2.3. Intervention Protocol

Before the experiment, participants were asked not to engage in any strenuous activity for at least 30 min. During this time, participants filled out the informed consent form. Subsequently, each participant performed their assigned walking condition according to randomization, wearing an HR monitoring wearable device.

To create the INVR condition, the same walking path as the OUT condition (Figure 2) was recorded using the Samsung New Gear 360 to produce a 360-degree video. During the INVR sessions, participants wore VR headsets and headphones, with a Samsung Galaxy S20 FE phone placed inside the headsets, providing them with an immersive virtual reality experience of the recorded outdoor walking path. The 360-degree video was recorded at the same speed of 4.5 km/h used for indoor sessions. This speed was chosen based on previous studies that used a similar setting [23].

The OUT condition was performed by walking outside on a coastal-view pathway (Figure 2). Each participant walked for 6 min at a predetermined speed of 4.5 km/h.

The IN condition was performed on a treadmill at 4.5 km/h speed. No headphones or other equipment were worn during this session, except for the HR monitor.

Immediately after the end of each walking condition, which lasted 6 min each, participants were measured for HR (average and maximum) and subjective ratings of enjoyment and mindfulness by answering the Physical Activity Enjoyment Scale (PACES-it) [25] and State of Mindfulness Scale for Physical Activity (SMS-PA) [26] questionnaires, respectively.

### 2.4. Measures

#### 2.4.1. Heart Rate

During the walking sessions, the participants’ heart rate was continuously monitored using a wearable device (Polar^®^ Ignite 2; Polar Electro Oy: Kempele, Finland), worn on the left wrist. After each test, the average HR (HR_mean_) and maximum HR (HR_max_) were recorded. Previous studies [27,28] have demonstrated the validity of this type of tracker for accurately assessing heart rate in adults.

#### 2.4.2. Physical Activity of Enjoyment Scale—Italian Version (PACES-it)

The Physical Activity Enjoyment Scale (PACES) is a questionnaire that measures an individual’s subjective enjoyment of PA [25,29]. It consists of 16 items with scores given on a 5-point Likert scale, from 1 (completely disagree) to 5 (completely agree): 9 items are positive (for example: “it energizes me”) and 7 items are negative (for example “It’s boring”) (Cronbach alpha 0.78 to 0.89) [30]. The PACES assesses various dimensions of enjoyment, including positive affect, psychological engagement and satisfaction with the activity [31,32]. It is a widely used tool in research and helps researchers understand individuals’ perceptions and attitudes toward PA, providing insights into the motivational factors that influence exercise behavior. PACES-It was administrated to the participants at the end of each session.

#### 2.4.3. State of Mindfulness Scale for Physical Activity (SMS-PA)

The State Mindfulness Scale for Physical Activity (SMS-PA) is an adapted version of the State Mindfulness Scale, focusing specifically on mindfulness during PA [33]. It was developed to capture the breadth of physical experiences during PA that were not adequately captured by the original scale. The SMS-PA measures the extent to which individuals attend to their physical exertion, muscular engagement, and bodily movements during PA. This scale consists of 12 items, with 6 items assessing mindfulness of the mind (thoughts and emotions) and 6 items assessing mindfulness of the body (movement, body sensations, muscle engagement). After each condition, the participants rate their agreement with each item on a 5-point scale ranging from 0 to 4, indicating the level of mindfulness experienced.

The SMS-PA is applicable for use with youth aged ten and older, and adaptations in Italian have also been developed (Cronbach alpha 0.85 to 0.90) [26]. It is intended to be completed immediately following participation in PA, providing insights into individuals’ mindfulness experiences during that specific activity.

### 2.5. Statistical Analysis

Statistical analyses were conducted using the JASP software v. 0.17.2.1 (JASP Team, 2023; jasp-stats.org). Data were presented as mean (M) values and standard deviations (SD) and were checked for assumptions of sphericity via Mauchly’s test. If the sphericity assumption was violated, the Greenhouse–Geisser correction was used.

The Shapiro–Wilk test was used to test the normality of all variables. One-way ANOVA with repeated measures was applied to detect any differences between the three conditions. If there was a significant difference between the conditions, then a post hoc test with Bonferroni’s correction was conducted to identify the significant comparison.

Eta squared (η^2^) was used to estimate the magnitude of the difference within groups and defined as follows: small: η^2^ < 0.06, moderate: 0.06 ≤ η^2^ < 0.14, and large: η^2^ ≥ 0.14 effect size (ES). Cohen’s *d* ES was calculated for the post hoc tests. The criteria to interpret the magnitude of Cohen’s *d* were as follows: small: 0.20 ≤ *d* < 0.50, moderate: 0.50 ≤ *d* < 0.79, and large: *d* ≥ 0.80 ES [34]. The statistical significance level was set a priori at *p* ≤ 0.05.

## 3. Results

All twenty-six participants who took part in the study were subjected to all three walking conditions and none of them reported injuries throughout the duration of the research. Table 1 shows all the changes experienced by participants between the three walking conditions.

One-way ANOVA with repeated measures found significant “within-subjects effects” for all the outcomes measures: HR_mean_ (F = 10.456, *p* < 0.001, η^2^ = 0.295, large ES), HR_max_ (F = 4.048, *p* = 0.035, η^2^ = 0.139, moderate ES), PACES-It (F = 21.861, *p* < 0.001, η^2^ = 0.467, large ES), SMS-PA (F = 5.345, *p* = 0.008, η^2^ = 0.176, large ES). Mauchly’s test of sphericity for HR_mean_ and HR_max_ indicated that the assumption of sphericity was violated (*p* < 0.05), and thus Greenhouse–Geisser correction was used.

Bonferroni’s post hoc test showed that HR_mean_ was significantly higher during the OUT compared to IN (t = −3.934, *p* < 0.001, d = 0.881, large ES) and INVR (t = −3.986, *p* < 0.001, d = 0.893, large ES) sessions (Figure 3).

Greater HR_max_ was found in the OUT than IN (t = −2.791, *p* = 0.022, d = 0.654, moderate ES) sessions. HR_max_ was not significantly different in OUT compared to the INVR (*p* > 0.05) session (Figure 4).

The level of enjoyment measured by PACES-It was also significantly higher in the OUT session compared to the others (OUT vs. IN: t = −5.452, *p* < 0.001, d = 1.251, large ES; OUT vs. INVR: t = −5.966, *p* < 0.001, d = 1.369, large ES) (Figure 5).

Greater SMS-PA scores were found in the OUT than IN sessions (t = −3.143, *p* = 0.008, d = 0.589, moderate ES). SMS-PA score was not significantly different in OUT compared to the INVR (*p* > 0.05) session (Figure 6).

## 4. Discussion

The study embarked on a meticulous exploration, probing the intricate impacts of three distinct walking conditions—namely INVR (indoor walking with virtual reality), OUT (outdoor coastal walking) and IN (indoor walking)—on psychological and physiological measures within a cohort of healthy adults. This endeavor aimed to unravel nuanced differences in heart rate (HR), enjoyment levels, and mindfulness experiences engendered by these diverse walking scenarios. The primary hypothesis postulated that the INVR condition would manifest values akin to the OUT condition across all variables under scrutiny. This hypothesis was substantiated, albeit selectively, finding confirmation in the case of HRmax and SMS measures. Intriguingly, no statistically significant differences were unearthed between the INVR and OUT conditions in these particular facets, signifying a degree of physiological equivalence.

Conversely, when we delve into HRmean and PACES measures, a different narrative emerges. These metrics exhibited lower values in both the IN and INVR conditions in comparison to the OUT condition. This highlights a palpable distinction in cardiovascular and experiential dimensions when engaging in indoor as opposed to outdoor walking. Participants showcased markedly higher HRmean values during OUT, lending credence to the notion that the natural outdoor environment poses distinctive physical demands, culminating in heightened cardiovascular exertion during outdoor ambulation. This is an observation that echoes extant research [35,36,37], reiterating the unique physiological implications of traversing natural terrains.

However, the absence of significant differences in HRmax between INVR and OUT introduces a compelling dimension to the discourse. It implies that the immersive virtual reality experience, an emblem of cutting-edge technology, can, to some extent, emulate the physiological responses induced by outdoor walking. While this substantiates the idea that technology-mediated indoor activities can approximate the physiological benefits of outdoor endeavors, distinctions were indeed detected between OUT and IN. The greater cardiovascular exertion associated with outdoor environments [35] became manifest in the higher value of OUT, reinforcing the irreplaceable facets of natural settings in PA.

Transitioning from the physiological to the experiential, enjoyment levels emerged as a pivotal parameter. IN and INVR were consistently reported as less enjoyable compared to OUT. This underscores a crucial psychological facet; participants derived heightened pleasure and satisfaction from the natural outdoor setting of OUT, indicating a potential intrinsic motivation embedded in outdoor activities. The immersive virtual reality experience during INVR, and the standard indoor environment of IN, were perceived as less enjoyable, potentially influencing motivation and adherence to PA. This aligns seamlessly with the findings of a parallel study [23], which reported significantly higher enjoyment during outdoor walking compared to indoor walking sessions with VR. Consequently, our study reinforces the intrinsic allure of the natural outdoor setting of OUT, postulating it as more enjoyable and, by extension, more conducive to sustaining PA over time.

Acknowledging the documented mindfulness benefits of self-paced outdoor walking [38,39,40] and the profound impact of VR-based exercise on mindfulness [41,42], our study aligns with these precepts. Mindfulness experiences showed no significant differences between OUT and INVR, suggesting that both environments facilitated a comparable state of mindfulness. This implies that the immersive virtual reality experience of INVR effectively engendered mindfulness, mirroring the serene and natural ambiance of coastal walking outdoors.

This study represents a noteworthy stride in unraveling the multifaceted dynamics of different walking conditions on both physiological and psychological facets. The confirmation of certain hypotheses, such as the physiological equivalence between INVR and OUT in specific parameters, is intriguing and opens avenues for further exploration. The consistent theme of outdoor walking being more enjoyable aligns with broader trends in PA research, emphasizing the pivotal role of natural environments in promoting sustained engagement. While the study, like any scientific endeavor, is not without limitations, it provides a robust foundation for future research endeavors that can build upon these insights, refining our understanding of how the choice of walking environment intertwines with the intricate tapestry of human health and well-being.

Expanding on the broader implications of this research, the study fundamentally underscores the need for a nuanced understanding of the interplay between technology-mediated indoor PA and the irreplaceable allure of outdoor environments. In an era dominated by virtual experiences and technology-driven leisure, the study offers a critical lens for the potential of immersive virtual reality in approximating the physiological responses and mindfulness benefits associated with outdoor walking. The findings suggest that, while technology can emulate certain aspects of the outdoor experience, the intrinsic joy and satisfaction derived from natural settings remain unparalleled.

The implications of these findings extend beyond the realms of academic inquiry into the practical domains of public health and well-being. Understanding the psychological and physiological nuances of different walking conditions can inform the design and implementation of interventions aimed at promoting PA. For instance, individuals constrained by factors such as inclement weather, lack of access, or time limitations may find a viable alternative in immersive virtual reality experiences. However, the study also cautions against a one-size-fits-all approach, highlighting the superior enjoyment associated with outdoor walking [23]. Thus, urban planning, workplace wellness programs, and health policies should consider the role of outdoor spaces in fostering PA and mental well-being.

Moreover, the study sheds light on the importance of mindfulness in the context of PA. The comparable mindfulness experiences between outdoor walking and virtual reality-based indoor walking suggest that technology, when designed with a mindful intent, can contribute to mental well-being. This insight is particularly relevant in a society grappling with sedentary lifestyles and stress-related health issues. Integrating mindfulness practices into technology-mediated PA could present a holistic approach to health promotion [22,26].

The longitudinal implications of different walking conditions constitute another area ripe for exploration. While the study provides a snapshot of acute effects, understanding the sustained impact of outdoor walking, indoor walking, and virtual reality-based activities can inform more robust recommendations for individuals and communities. Longitudinal studies tracking participants over extended periods could elucidate the enduring benefits and potential habituation to different walking modalities.

Finally, this study navigates the intersection of technology, PA, and well-being, unraveling layers of complexity in how different walking conditions shape our physiological responses, enjoyment levels, and mindfulness experiences. As society grapples with evolving patterns of PA and increasing reliance on technology, these insights become pivotal signposts. They guide us in harnessing the potential of immersive virtual reality for health promotion while underscoring the timeless allure and benefits of natural outdoor environments. The study, therefore, beckons further exploration, inviting researchers, policymakers, and practitioners to embark on a journey of deeper understanding and innovative interventions at the nexus of human movement and well-being.

### Strengths and Limitations

The study’s reliance on a relatively small sample size, drawn exclusively from university sports science students, introduces a potential source of bias, limiting the generalizability of the results to a broader population. The predominantly homogeneous participant cohort may not fully represent diverse demographic groups, affecting the external validity of the findings. While the cross-over design is pragmatic for short-term analyses, it complicates the observation of long-term effects, making it challenging to draw conclusions about sustained impacts over time. The novelty associated with participants’ first-time use of the visor may have induced emotional arousal, potentially influencing physiological parameters. The study recognizes this as a potential confounding factor. Due to the specific conditions of the study and the unique sample, caution is warranted in generalizing the findings to broader populations or different settings. The study, by incorporating virtual reality technology, might introduce a bias toward technology-mediated activities, and the findings may not fully capture the preferences and responses of individuals less accustomed to such technology. The study, while providing valuable insights, may lack a real-world context. Participants’ experiences in a controlled study environment might differ from their experiences in their daily lives. The study primarily focuses on acute effects, and while this provides a snapshot, it may not fully capture the sustained impact and habituation to different walking modalities over an extended period. The study acknowledges the potential for participant bias due to the small and specific sample, emphasizing the need for future research with broader participant diversity. The caution against a one-size-fits-all approach, while valid, adds complexity to the applicability of the study’s findings to diverse populations and contexts.

On the other hand, during in this study a meticulous exploration of three distinct walking conditions was conducted, providing a detailed analysis of their impacts on psychological and physiological measures. The research question addressed a pertinent issue, examining the effects of different walking conditions on both physiological (heart rate) and psychological (enjoyment, mindfulness) aspects, crucial for overall well-being. The inclusion of virtual reality (INVR) as one of the walking conditions adds innovation to the study, reflecting contemporary trends in technology and its potential impact on PA and well-being. The study confirmed certain hypotheses, such as the physiological equivalence between INVR and OUT in specific parameters, contributing valuable insights to the understanding of how technology-mediated indoor activities compare to outdoor experiences. It contributes substantially to the existing body of knowledge on the interplay between walking conditions and human well-being, emphasizing the importance of natural environments in promoting sustained engagement. The study not only focused on physiological measures but also delved into psychological factors such as enjoyment and mindfulness, providing a holistic view of the impact of different walking conditions. The cross-over design offered insight into the short-term effects of different walking conditions. With this study design, the influence of confounding variables was reduced because each subject acted as his or her own control; moreover, it produced rapid responses to the research question because counterbalanced randomization could show cause and effect [43]. Thus, our study sought to bring novelty to the field of research.

Physical activity with virtual reality (VR) can serve as a compelling alternative to outdoor physical activity, especially in circumstances where outdoor engagement is challenging or limited. VR offers an immersive and interactive experience that simulates outdoor environments, providing users with a dynamic and engaging workout. This alternative is particularly beneficial in adverse weather conditions, urban settings with limited green spaces, or situations where individuals face time constraints. Moreover, VR can cater to diverse preferences by offering various virtual landscapes and activities, making it adaptable to different fitness levels and interests. Incorporating gamification elements further enhances motivation, making VR-based physical activity an appealing substitute for outdoor exercises. However, it is essential to balance this with the understanding that the intrinsic benefits of natural settings cannot be entirely replaced. Therefore, the use of VR should be strategic, considering individual preferences, accessibility, and the overarching goal of promoting sustained physical activity and well-being.

Finally, while the study makes significant strides in unraveling the dynamics of different walking conditions, it is crucial to interpret its findings within the context of these strengths and limitations. Future research should aim to address these limitations for a more comprehensive understanding of the interplay between technology-mediated indoor activities and the allure of outdoor environments on human well-being.

## 5. Conclusions

To the best of our knowledge, this is the first study that provides new insights into the physiological and psychological effects of different walking conditions, including those in coastal environments. OUT has greater HR_mean_ values with respect to INVR and IN, but HR_max_ did not differ significantly between OUT and INVR. This suggests that PA in an immersive environment may lead to physiological loads comparable to the outdoor setting. OUT emerges as an enjoyable and engaging alternative to IN and INVR. Also, the immersive virtual reality experience of INVR presents a comparable mindfulness response to the OUT, supporting its potential use in PA interventions. Thus, both OUT and INVR offer similar levels of mindfulness experiences, highlighting the benefits of incorporating outdoor coastal walking and virtual reality-enhanced indoor walking in PA interventions. These findings imply the need to provide diverse and stimulating environments to enhance enjoyment and motivation for PA and improve individual well-being and health.

## Figures and Tables

**Figure 1 jfmk-09-00011-f001:**
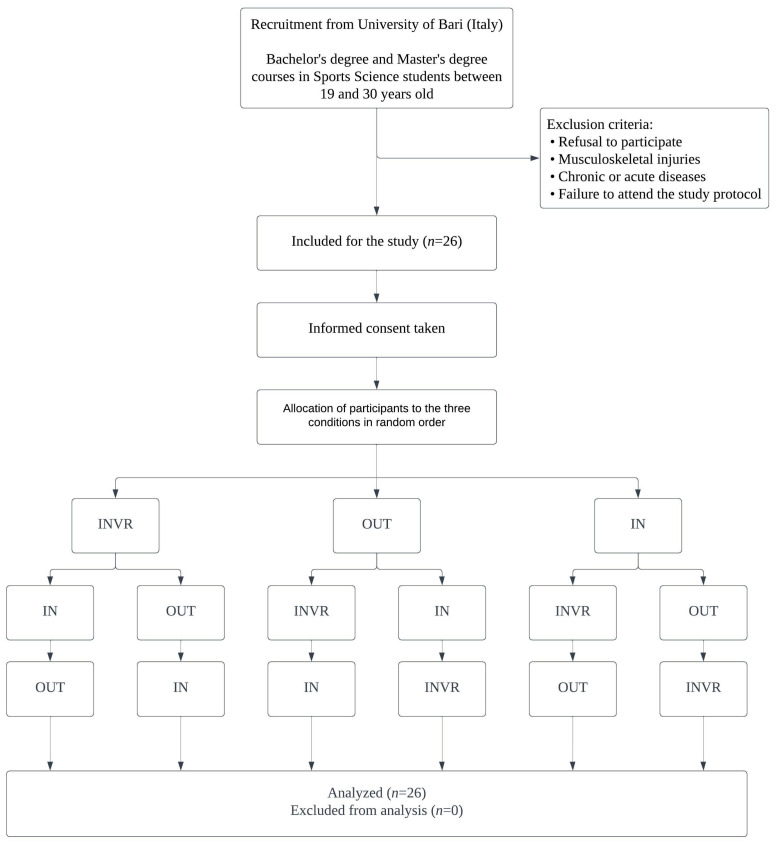
Randomized allocation of participants to the three conditions (indoor walking with virtual reality (INVR), outdoor coastal walking (OUT), standard indoor walking (IN)).

**Figure 2 jfmk-09-00011-f002:**
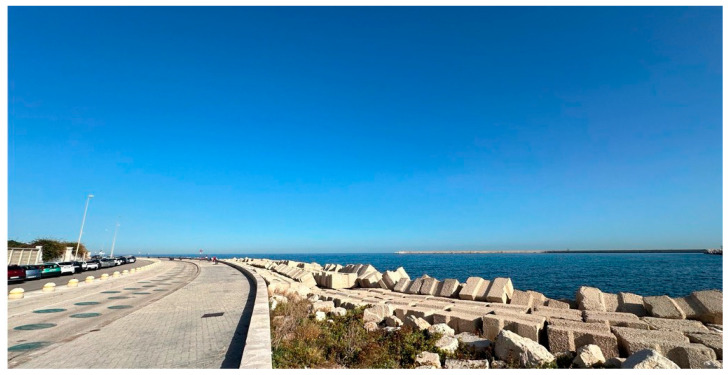
Coastal pathway used for the outdoor coastal walking (OUT) and indoor walking with virtual reality (INVR) sessions.

**Figure 3 jfmk-09-00011-f003:**
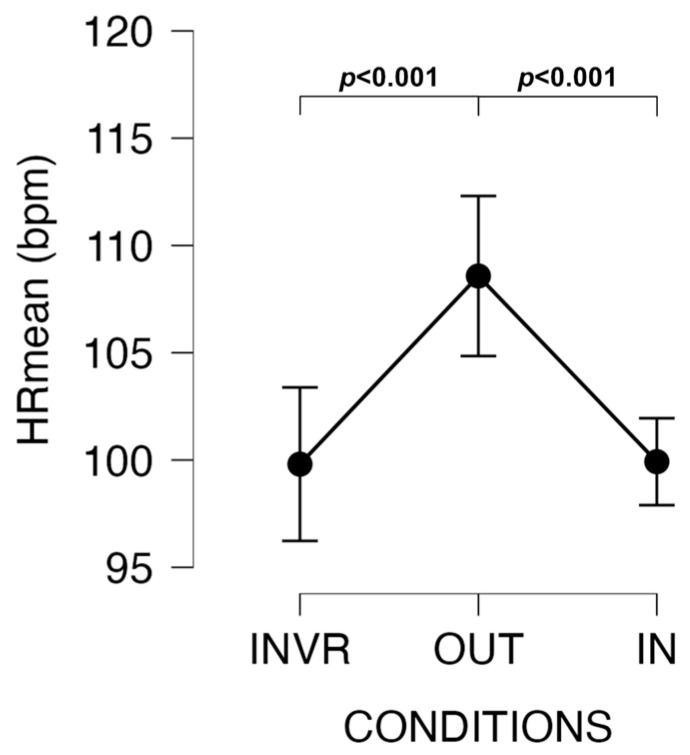
Changes in HRmean among conditions.

**Figure 4 jfmk-09-00011-f004:**
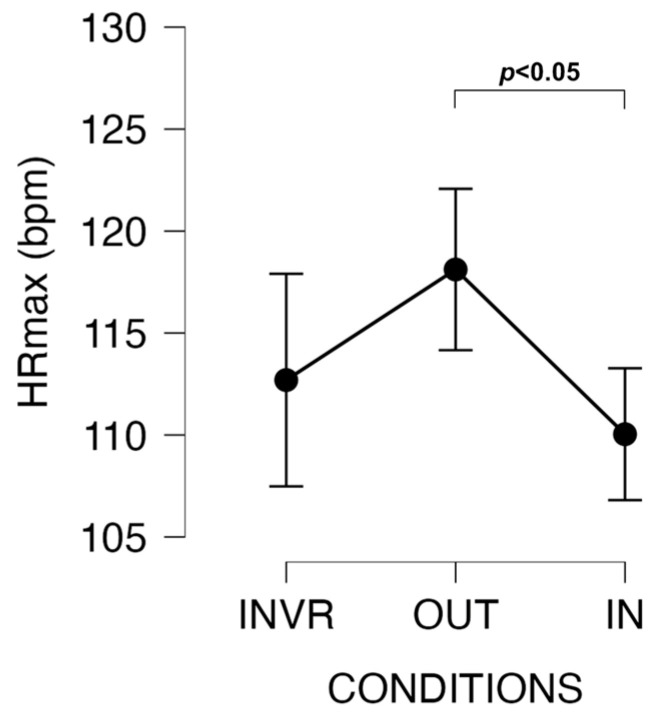
Changes in HRmax among conditions.

**Figure 5 jfmk-09-00011-f005:**
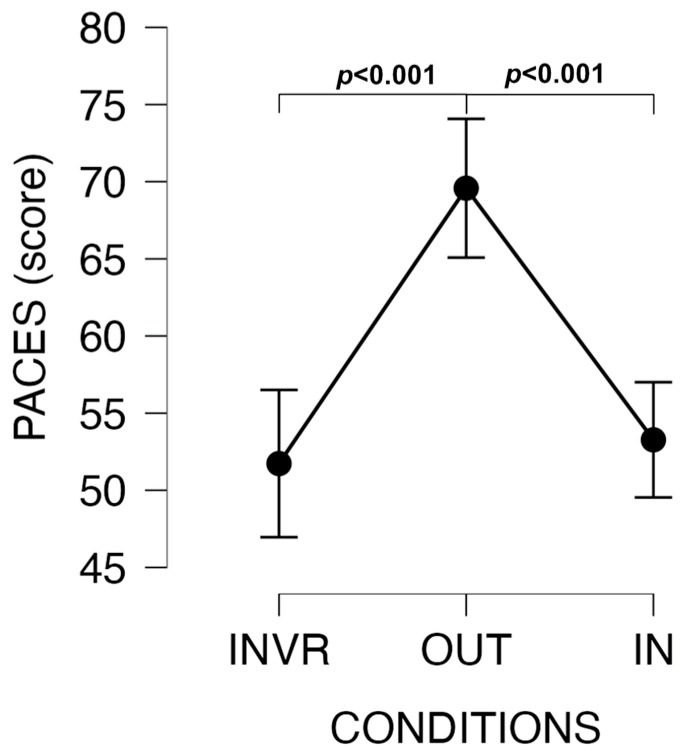
Changes in PACES scores among conditions.

**Figure 6 jfmk-09-00011-f006:**
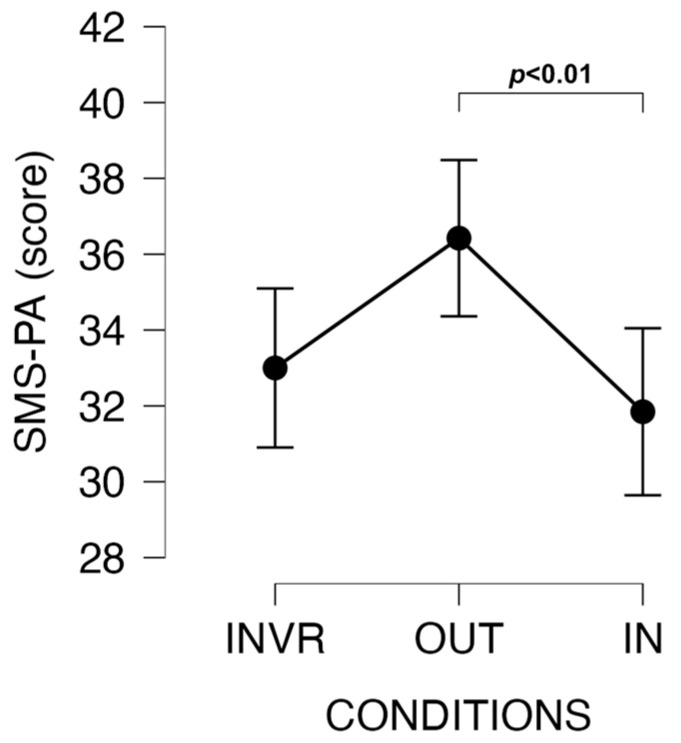
Changes in SMS-PA scores among conditions.

**Table 1 jfmk-09-00011-t001:** Changes found among the three walking conditions.

Variables	INVR	OUT	IN
HR mean (bpm)	99.8 ± 12.4 ^a^***	108.6 ± 7.7 ^a^***^,b^***	99.9 ± 8.8 ^b^***
HR max (bpm)	112.7 ± 17.4	118.1 ± 8.5 ^b^*	110.0 ± 9.0 ^b^*
PACES-It (scores)	51.7 ± 17.2 ^a^***	69.6 ± 7.8 ^a^***^,b^***	53.3 ± 12.3 ^b^***
SMS-PA (scores)	33.0 ± 9.1	36.4 ± 7.8 ^b^**	31.9 ± 6.1 ^b^**

Data are reported as mean ± SD. Abbreviations: INVR, Indoor Walking with Virtual Reality; OUT, Outdoor Walking; IN, Indoor Walking; HR, Heart Rate; PACES, Physical ACtivity Enjoyment Scale; SMS-PA, State of Mindfulness Scale for Physical Activity. ^a^ Significant difference between INVR and OUT; ^b^ significant difference between OUT and IN. * *p* < 0.05; ** *p* < 0.01; *** *p* < 0.001.

## Data Availability

The data presented in this study are available on request from the first author.

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
