# Peer review of "Impact of Coastal Walking Outdoors and Virtual Reality Indoor Walking on Heart Rate, Enjoyment Levels and Mindfulness Experiences in Healthy Adults"

_jfmk, 2023, doi:10.3390/jfmk9010011_

Round 1

Reviewer 1 Report

Comments and Suggestions for Authors

Basic reporting

The manuscript is generally well-written and easy to read. Although the results of the study are interesting and the overall structure and content seem robust, I have some suggestions that the authors need to address to improve the quality of their work. 

Introduction

The literature on the subject is sufficiently well summarised.

Considering the multiple repetitions of the term physical activity, I suggest rephrasing some sentences and shortening the term to PA the first time it is used.

Line 81-83: Please rephrase this sentence, it could be clearer.

The term "INVR condition" is introduced without prior explanation (the abstract is considered a stand-alone section, so even if you have defined what it means, it needs to be explained again in the introduction section). Ensure that any specific terminology or abbreviations are defined or explained when first introduced to enhance reader understanding.

The text discusses augmented reality (AR) and virtual reality (VR) but does not explicitly define or differentiate between them. Providing a brief explanation of these technologies might be helpful for readers who may not be familiar with them.

Methods

The methods section is sufficiently well described.

The intervention protocol is well-described. However, you may want to consider providing additional information on the rationale behind the choice of the walking speed (4.5 km/h) and the duration of 6 minutes for each condition. Furthermore, you mentioned that assessments and conditions were conducted between 9 a.m. and 12 a.m. Consider briefly discussing why this time frame was chosen.

Validity of the findings

The results and discussion section are quite clear and organised. The parameters considered are well presented.

When referencing previous research (e.g., [32–34]), could be useful to briefly summarize the key findings from these studies to provide context for readers who may not be familiar with them.

I’m not sure and this is a personal consideration: you mention the novelty of using this technology as a possible bias. Was every participant new to this technology or had some already used it? Maybe, even familiarity with this technology could be a confounding factor. If yes, could be useful state it.

Reviewer 2 Report

Comments and Suggestions for Authors

The authors exhibit commendable honesty in acknowledging the scarcity of studies that could provide further data.

The role of the editors becomes pivotal in this scenario. I reiterate, the quality of the work is exceptional.

Author Response

Dear Reviewer,

thank you so much for your appreciation.

Reviewer 3 Report

Comments and Suggestions for Authors

1.      Consider augmenting the introduction with more specific data and research results concerning the surge in sedentary behavior and its associated health issues. This addition will help underscore the urgency of the problem.

2.      Elaborate on the rationale for selecting participants from the bachelor's and master's degree courses in Sports Science. Clarify whether these participants represent a sedentary population, and discuss potential biases introduced by using this specific group for the experiment.

3.      Provide a more detailed explanation of why a crossover randomized controlled study design was chosen and its applicability to the research question. This additional information would enhance understanding regarding the methodological choices of the study.

4.      While the discussion mentions some physiological and psychological significance of the results (e.g., differences in heart rate and similarity in mindfulness experiences), it is suggested to further emphasize the potential practical implications of these findings on individual health and well-being. Connecting the results to real-world applications would strengthen the discussion.

Round 2

Reviewer 3 Report

Comments and Suggestions for Authors

The quality of this paper was improved.